# Early Saphenous Vein Graft Aneurysm Rupture: A Not So-Late Complication. Case Report and Comprehensive Literature Review

**DOI:** 10.3390/biomedicines11010220

**Published:** 2023-01-14

**Authors:** Eleonora Mezzetti, Aniello Maiese, Federica Spina, Fabio Del Duca, Alessandra De Matteis, Marco Di Paolo, Raffaele La Russa, Emanuela Turillazzi, Vittorio Fineschi

**Affiliations:** 1Department of Surgical, Medical and Molecular Pathology and Critical Care Medicine, Section of Legal Medicine, University of Pisa, 56126 Pisa, Italy; 2Department of Anatomical, Histological, Forensic and Orthopedical Sciences, Sapienza University of Rome, Viale Regina Elena 336, 00161 Rome, Italy; 3Department of Clinical and Experimental Medicine, University of Foggia, 71122 Foggia, Italy

**Keywords:** saphenous vein graft, rupture, bypass

## Abstract

Background and Objectives: Saphenous vein graft (SVG) is a cardiac surgical practice used to create a cardiac bypass in cases of coronary artery obstruction. It consists of a surgical procedure that involves the creation of an aorto-coronary communication by a venous conduit (saphenous vein) to bypass coronary stenosis and allow cardiac revascularization. This practice can be affected by early and late complications. The most feared complication is graft aneurysm or pseudoaneurysm degeneration and rupture which are considered late complications. This paper presents a rare case of SVG aneurysmal rupture that occurred 24 h after surgery and a review of the literature to provide a general look at the state of knowledge. Materials and Methods: The systematic review was carried out using the guidelines according to the PRISMA method. Results: Cases of aneurysmal rupture have never been described prior to one month after surgery. The male sex and subjects under 45 are the most affected by this complication. Death occurs in less than half of the cases, being more frequent in young people. Performing a CT or angio-CT examination led to the diagnosis. Conclusions: It is impossible to estimate the implanted vessel’s quality, so postoperative follow-up is fundamental. Transesophageal ultrasound can be useful, and hematochemical tests are valuable early diagnostic tools, whrease CT and angio-CT can be useful even months after surgery. Forensic analysis should always perform an autopsy and graft histological examination.

## 1. Introduction

Coronary artery bypass grafting (CABG) is a surgical technique that involves the placement of one or more arterial or venous grafts to bypass coronary circulation. The graft bypasses the coronary artery blockage, which is the result of atherosclerotic disease, for myocardial revascularization. The saphenous vein is often used as a venous graft; however, this technique has high failure rates, ranging from 10 to 25 percent in the first 12–18 months [1] compared to arterial grafts. Moreover, this technique is recommended when there is a high degree of obstruction in one of the major coronary arteries or if the percutaneous coronary intervention (PCI) failed in patients with angina refractory to medical therapy and PCI [2]. Aneurysmal dilatation and pseudoaneurysm degeneration of SVGs are unusual but potentially fatal complications rarely described in the scientific literature [3,4]. Their rupture occurs in only 8% of cases with catastrophic complications [5] such as rupture and death from cardiac tamponade or shock. This article presents a rare case of hemorrhagic shock due to a rupture of an SVG pseudoaneurysm one day after the implant. In addition, a systematic review involving the ruptured SVG aneurysms or pseudo aneurysms has been performed. This work aims to highlight the timing of the presentation of these complications, to help clinicians in a correct and timely diagnosis, and to prevent fatal outcomes. Furthermore, this review could encourage the creation of diagnostic-therapeutic protocols with specific peri and post-operative control tests in all patients with saphenous venous grafts.

## 2. Case Report

A 53-year-old man recovered after a syncopal episode. At the hospital, control cardiac CT showed a severe calcific stenotic degeneration of the aortic valve with bicuspid morphology. An ectasia of the ascending aorta and the proximal tract of the left subclavian artery was also diagnosed. The clinical situation required surgical treatment for valve replacement. The procedure consisted of replacing the ascending aorta, aortic root, and aortic valve with bioprothesis (Edwards Inspiris Resilia 29 mm) according to the Bentall- De Bono procedure. This type of implant consisted of a biological valve tube composed of a Vascutek Valsalva prosthesis (32 mm) joined to an Inspiris Resilia valve prosthesis (29 mm). Coronaries were manually implanted directly onto the biological prosthesis by continuous double sutures. During the procedure, the patient suffered a sudden episode of ventricular fibrillation (VF). The physicians hypothesized that myocardial hypoperfusion was caused by ostial occlusion, secondary to double suture-induced lumen reduction, and occurred during valve implantation. After a new sternotomy, the sanitarians performed immediate cardiac massage and heparinization.

Considering the complexity of the inserted prosthesis, surgeons could not proceed with an aortotomy, so a double CABG was performed. The type of bypass performed was decided based on what was previously agreed with the patient in the “informed consent” in case of an ischemic emergency.

A double bypass was performed to the anterior descending artery and the right marginal artery, a branch of the right coronary artery, by autologous saphenous vein graft. SVG quality and continence were tested by the administration of a heparinized saline solution. After only four hours, the hemoglobin level began to drop. An echocardiography performed 7 h after the surgery showed a small amount of pericardial effusion. The day after, the patient’s clinical condition became critical; no blood transfusion or epinephrine injections could prevent hemorrhagic shock and death. The autopsy revealed a rupture of a pseudoaneurysmatic degeneration of SVG packaged in the main body of the anterior descending artery graft. A secondary massive blood leakage in the pericardium was found (Figure 1).

Pericardium appeared vastly infiltrated and full of clots. Pleural cavities contained massive quantities of blood:830 ccs on the right side and 230 ccs on the left one. Histological examination showed that the fissure of the SVG (Figure 2) occurred first in the deepest layer and spread to the more superficial layers. Specifically, at the rupture, the preparation showed infiltration by red blood cells and fibrin. In contrast, the rest of the vessel appeared normal, in the absence of pathologic changes in the vassal tonaca or alterations in the small collateral vessels. Autopsy investigation allowed us to assess the cause of death: hemorrhagic shock caused by the SVG’s pseudoaneurysm rupture.

Autopsy examination also showed a reduction in the ostia lumens of the implanted coronaries due to the use of a double suture, with more evidence at the left coronary ostium. Subsequent studies allowed attributing the VF episode to this defect in implantation that reducing the lumen induced an episode of critical ischemia.

## 3. Materials and Methods

In addition to the previous case report, a systematic review was carried out according to the Preferred Reporting Items for Systematic Review (PRISMA) standards 1. A systematic literature search and a critical appraisal of the collected studies were conducted. An electronic search of PubMed, Science Direct Scopus, Google Scholar, and Excerpta Medica Database (EMBASE) from the inception of these databases to October 2022 was performed. The search terms were “Saphenous Vein Graft Pseudoaneurysm Rupture”, “Saphenous Vein Graft Aneurysm Rupture”, “Saphenous Vein Graft Aneurysm + autopsy”, “Saphenous Vein Graft Pseudoaneurysm + autopsy”. The bibliographies of all located papers were examined and cross-referenced for further relevant literature. The methodological appraisal of each study was conducted according to the PRISMA standards, including the evaluation of bias. Data collection entailed study selection and data extraction. The following inclusion criteria were used: (1) original research articles, (2) reviews and mini-reviews, (3) case reports/series, (4) only papers written in English.

Two researchers (E.M. and F.S.) independently examined the papers with title or abstracts that appeared to be relevant and selected the ones that analyzed the ruptures of an SVG aneurysm or pseudoaneurysm. Disagreements concerning eligibility between the researchers were resolved by a consensus process. No unpublished, pre-print, or grey literature was searched. Data extraction was performed by two investigators (A.M. and M.D.P.) and verified by two other investigators (F.D.L. and A.D.M.). This study was exempt from institutional review board approval as it did not involve human subjects.

## 4. Results

A total of 138 articles were identified, from which duplicates, articles without explicit mention of ethical rights, and study designs were removed. These publications were carefully evaluated considering the main goals of the review. Only articles written in English were then considered. The final meta-analysis included a total of 23 articles and scientific papers comprising original research articles, case reports, and case series. Figure 3 illustrates our search strategy.

As shown in Table 1, the patients’ sex, age, and time were identified such as the mode of aneurysmal rupture, symptoms, and outcome. In cases of aneurysmal rupture, the lesion was always preceded by dilatation of the venous graft. Rhiai et al. [6] and Benchimol et al. [7] present two cases of asymptomatic aneurismal rupture such as Salcedo (2013). 

Douglas et al. [8] present a case of mycotic aneurysm rupture, whereas Foster et al. [9] discuss an aneurysmal rupture that occurred intraoperatively. Aneurysmal and pseudoaneurysmal lesions are generally more common in the male gender (19/23) than in the female gender (4/23), albeit with a lower number of deceased subjects (Table 2). Among women, the percentage of pseudoaneurysms is the same as that of aneurysms (2/4 respectively), as well as deceased versus surviving subjects. There are 16 cases of aneurysmal and pseudoaneurysmal ruptures in subjects over 45 years old, whereas among subjects under 45 years old, only 7 cases were described. In the male group, most ruptures were diagnosed 5 years after implantation, while in women it happens more often in the first 5 years (2/4 cases). In only two cases was the time to rupture not determined. As a diagnostic method, computed tomography was used in 14/23 cases and chest X-ray was used in 9/23 cases; examinations such as angiography or aortography were performed in 7/23 cases and 5/23 cases, respectively. In only two cases, cardiac tamponade or cardiogenic shock is described (1/2), while chest pain appears to be the most frequent symptom (11/23 cases).

Among the male subjects, only one was found to have diabetes, while hypertensive disease was recorded in four cases. Most of the subjects in both the male and female groups showed a predilection for vascular disease (Table 3).

Table 4 shows the main break-up times in subjects above and below 45 years of age. In both groups, rupture occurs more frequently after 5 years (17.4% for those under 45, and 43.5%, for those over 45, respectively), and it is more frequent in older subjects (43.5% of cases).

Studying the population, individuals under the age of 45 have a higher mortality rate (17.4%) than individuals under the age of 45. Similar considerations can be made in the case of pseudoaneurysms. Among individuals with aneurysms, only 3/23 died because of rupture, whereas in individuals with pseudoaneurysms, death occurred in only one case (Table 5).

Autopsy examination was performed in 26.1% of cases, 17.4% in the case of the male group, and 8.7% for the female group (Table 6).

Internal examination showed one case of pericarditis and two cases of heart weight growth. In the other two cases, graft occlusion was diagnosed, whereas the occlusion was registered in only one case. Hemomediastinum occurred in only two cases of aneurism rupture, as hemopericardium in only one case. Left ventricle dilatation was registered in two cases, similar to heart weight growth. In cases of pseudoaneurysms, only occlusions or ruptures were recorded. At a histology examination, acute phlebitis and polymorphonuclear were found in two cases such as myocardial infarction and hemorrhage. In pseudoaneurysm, exclusively graft atherosclerosis was found (Table 7).

## 5. Discussion

In heart surgery, CABG is a procedure to bypass the stenotic lesion of the coronary artery using aortocoronary grafts (free arterial grafts or saphenous vein grafts) and internal thoracic arteries in situ with only a distal anastomosis in the coronary artery [28].

Murray studied the arterial bypass in 1953 by suturing mammalian, carotid, and axillary arteries to the heart [29]. In 1957 Smith et al., demonstrated that a segment of saphenous vein taken from a leg could be useful to create a bypass from the aorta to the myocardium, maintaining the same circulation granted by coronaries [30]. Grafts can be distinguished into autologous or heterologous, possibly used for implantation. However, the most important classification divides grafts into arterial and venous [31].

The latest 2018 guidelines on myocardial revascularization recommend preferring arterial conduits whenever possible, especially in young subjects [32]. In the end, venous or arterial grafts may be used in young subjects for bypass implantation.

Between arterial vessels, the arteria radialis, epigastric artery inferior, ulnar artery, left gastric artery, right gastroepiploic artery, splenic artery, arteria subscapularis, and circumflex femoral artery are the most selected grafts [33]. Traditional CABG remains the gold standard for multi-vessel coronary artery disease with complex (left anterior descending artery) stenosis [34].

Both types of grafts are subject to early and late complications. In the case of venous CABG, early complications are represented by venous occlusion, embolism, infections, arrhythmias, and cardiac arrest [35]. Long-term complications generally arise within 10–15 years after surgery and are based on mortality, absence of cardiac disease, and need for revascularization. Studies show that most cardiac vein graft complications occur within the first year after implantation (10–20%), while vein graft failures are between 5 and 10% of cases between 1 and 5 years [36]. Aneurysms or pseudoaneurysms were generally identified as late complications. The literature review showed numerous cases of an aneurysmal rupture occurring within five years after surgery. In this regard, Mohara et al. [25] described an SVG pseudoaneurysm rupture one month after surgery in a 73-year-old woman.

On the other hand, Douglas et al. [8] described an aneurysmal rupture that occurred seven days after surgery. Such a timely rupture was explained by the simultaneous mycotic graft infection that generated an aneurysm in the midportion of the SVG. Also, Montgomery et al. [19] described a surgical site infection with aneurysmal development; it is reasonable to think that aneurysmal degeneration of the graft, in this case, was a manifestation of the endocarditic process resulting from the surgical site infection. We can assume that infections, endocarditis, or septic processes can occur and result in the degeneration of the venous graft and, secondarily, in its rupture.

In our case report, an aneurysmal rupture occurred uncommonly in the first 7 h after surgery, when an echocardiographic examination showed a small amount of pericardial effusion. Among the cases listed in the review, no earlier cases of rupture are enumerated. It would be erroneous to continue to classify aneurysmal rupture as an exclusively late complication, although it rarely occurs within a few months or days of surgery. Certainly, aneurysmal rupture can occur early, even in the absence of infectious disease. Harskamp et al. [37] demonstrate that the main risk factors for aneurysmal rupture are included in the female sex [38], younger age [39], hypercholesterolemia [40], and previous heart failure or low ejection fraction.

In our study, most aneurysmal ruptures occur in subjects older than 45, with a higher survival rate overall than younger subjects. Moreover, the female sex appears less affected by this complication; unexpectedly, the surviving subjects turned out to be almost three times the deceased subjects. Goldman et al. [39] described dyslipidemia as among the most common risk factors for aneurysm development. In our study, only Kosky et al. [18] and Karwande et al. [21] showed a case of hyperlipidemia. Hypertension is the most common comorbidity, although in most cases, the subject is affected by vascular diseases such as severe aortic insufficiency, vasculitis, or myocardial infarction [27]. We also found that all subjects with aneurysmal rupture have diffuse atherosclerotic pathology. Obviously, atherosclerotic diseases of coronary circulation should always be considered a risk factor for aneurysm development because it is why venous bypass was packaged. Chronic obstructive pulmonary disease does not appear to be directly correlated with an increased risk of aneurysmal rupture. Diabetes is typically associated with an increased risk of graft occlusion, but it is not a common risk factor for aneurysm formation and rupture. However, it cannot be ruled out that diabetes may cause vascular degeneration such that ectasic degeneration is promoted [7]. Indeed, in the literature review presented, in two cases of aneurysmal rupture, the graft was also occluded, but patients did not result in having diabetes [15,24].

Clinically, when an aneurysmal rupture occurs, the symptomatology is usually hemorrhage, hemothorax, and hemorrhagic shock [19,23,27]. Chest pain is by far the most common symptom of aneurysmal rupture.

The most reliable diagnostic technique for the diagnosis of aneurysmal rupture is CT or Angio-CT examination [41,42,43]. On the other hand, coronary angiography confirms the diagnosis and determines the vein graft patency. Certainly, a chest X-ray or echocardiographic examination should be performed at the first instance. Transesophageal echocardiography retains a sensitivity and specificity of 99% and 98% for cardiac lesions [44]. It is a minimally invasive and useful examination for obtaining an early diagnosis. In our case report, performing transesophageal echography allowed the first visualization of pericardial effusion just seven hours after surgery. This finding is also supported by the literature review, which revealed four cases in which RX or CT was associated with ultrasonography [13,18,26,27]. Regardless, coronary angiography is strongly indicated even if the diagnosis was carried out through non-invasive techniques. In this regard, angiography allows the diagnosis of an aneurysmal lesion and the control of rupture by implanting medicated stents [45,46].

An important element in diagnosing aneurysmal rupture remains the autopsy and histologic examination. In our study, an autopsy examination was performed in only six cases. Histologic examination of the graft showed mostly other atherosclerotic actions, myocardial infarction, and polymorphonuclear infiltrate. Occasionally, fibrin and calcifications were detected, especially for late-onset aneurysmal lesions. These modifications certainly indicate a pathological process developed for some time; not surprisingly, the graft rupture occurred over five years [13,15]. Hassantash et al. [47] describe atherosclerosis and leukocyte infiltration as typical pathophysiological alterations. In our study, they are present in half of the cases.

In the case we presented, histological examination revealed sloughing between the inner and superficial layers of the graft. Initially, there was a rupture of the inner layer and then an injury of the outermost layer.

The presence of red blood cells and fibrin indicates that sloughing occurred before the rupture [48]. If the injury had been secondary to graft manipulation, the rupture of the two layers would have been simultaneous, or primarily in the outer layer, without evidence of sloughing and fibrotic organization. Only hemorrhagic infiltration would be documented without fibrotic organization in case of simultaneous rupture. Conversely, if rupture first occurs in the inner layer, coagulation processes will begin, and aneurysmal degeneration will occur simultaneously. Specifically, the formation of an aneurysm is favored by the high pressures of the arterial circulation in comparison to the venous one, which, as a result, will experience subsidence and dilatation [49].

Without a histological examination, it is difficult to determine whether the graft injury is attributable to iatrogenic manipulation or intrinsic atherosclerotic and microscopic changes in the graft. Even though bypass is packaged under emergency (24.7%) or elective conditions (70.8%) [50,51], an evaluation of vessel quality or continence by infusion of heparinized saline is always planned. However, this method does not allow us to understand whether the vessel is at risk for aneurysmal degeneration. In this case, surgeons can only estimate the risk of aneurysmal degeneration based on the patient’s risk factors. Certainly, systemic pathologies or the presence of varicose disease are tangible elements that can promote aneurysmal degeneration. In the case we presented, the patient did not have systemic pathology and did not have varicose degeneration of the saphenous vein at the time of surgery. The vessel was found to be of good quality and continence at the heparinized saline infusion test. Consequently, aneurysmal degeneration of venous vessels can often occur even in healthy structures. From this point of view, proper postoperative follow-up is the only way to diagnose aneurysmal rupture early.

Our case report presents a rare precocious aneurysmal rupture, in which follow-up hemoglobin values and transesophageal ultrasonography were two key tests for early diagnosis. Nevertheless, the shock condition set in within a short time, leading to the rapid death of the patient. The reduction in hemoglobin levels appeared compatible with usual postoperative anemia in major surgery [52]. The initial aneurysmal fissuring did not lead to immediate hemorrhagic shock, and the pericardial effusion was only evident during the transesophageal performance. From this perspective, it is important to emphasize the need to focus attention precisely on this or the need to pay more attention to clinical symptoms and to make serial diagnostic checks in the immediate postoperative CABG.

In conclusion, proper monitoring of patients with venous bypass should respect the performance of hematochemical tests and a postoperative transesophageal ultrasound. On the contrary, thoracic X-rays or CT are valid tests in the long term. Coronary angiography remains the gold standard, capable of detecting even minute changes.

This article shows the need for continued collaboration between the clinical and forensic worlds. Forensics can offer valuable help to clinicians in post-mortem diagnosis and in reducing adverse events. Indeed, good prevention based on proper clinical risk management, such as the present case, can greatly reduce early or late complications even when extremely rare.

## 6. Conclusions

This work reviewed 23 cases of SVG aneurysms and pseudo aneurysm rupture, accompanied by a specific case report dealing with an SVG pseudoaneurysm rupture. Our work is important for clinicians to recognize the possible early and late complications of bypass intervention. As our case shows, pseudoaneurysms, considered late complications, can also occur in extremely early circumstances from bypass intervention. In the suspicion of a possible rupture of the SVG, clinicians should perform serial checks based on hemoglobin determination and angio-CT exams. Currently, no useful instrumental investigations exist to identify the continence and integrity of the venous vessels selected after the bypass intervention. It would be useful to develop a correct diagnostic process, such as a serious follow-up of postoperative hemoglobin values and a transesophageal examination. CT or angiography are also appropriate tests to identify aneurysmal degeneration early. Special attention should be given to those patients with hypertension, diabetes, and vascular diseases.

In case of patient death, it is good practice to perform an autopsy examination to demonstrate aneurysmal degeneration of the graft. Forensic pathologists should perform careful autopsy examination and histological analysis, with attention to the venous graft, the aorta- in all its components-, and the coronaries. A histological investigation should be carried out on fragments of graft walls to analyze any inner layer lesions and differentiate spontaneous ruptures from iatrogenic layers. A typical finding of aneurysmal ruptures is the sloughing between the graft’s inner layer and the most superficial.

In conclusion, this review shows that, in the presence of a coronary aortic bypass intervention, in the immediate postoperative period, clinicians should perform serial biochemical-instrumental tests in consideration of the possible early complication of SVG rupture.

It is recommended for forensic pathologists to conduct a careful histological examination of the graft segment in which the rupture is present.

## Figures and Tables

**Figure 1 biomedicines-11-00220-f001:**
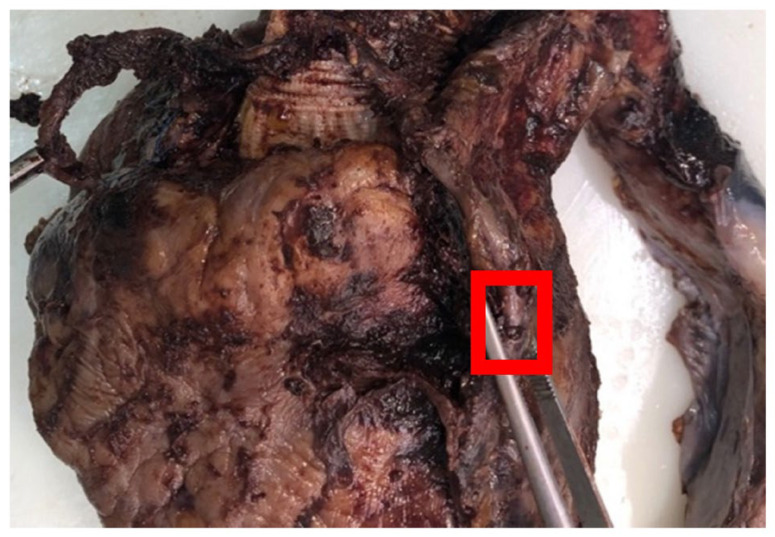
The left coronary graft site of pseudoaneurysm with wall rupture (red box).

**Figure 2 biomedicines-11-00220-f002:**
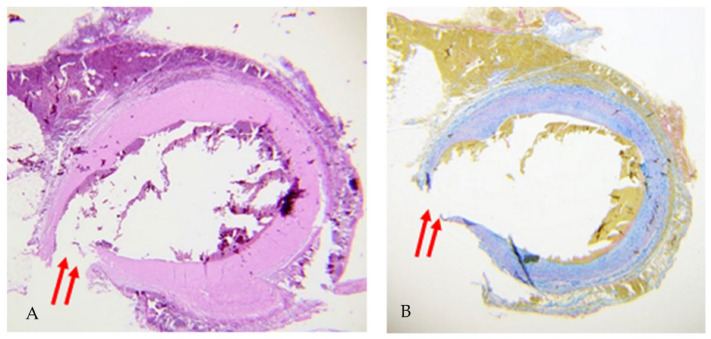
This figure shows two histological preparations, one stained with hematoxylin-eosin (**A**) and the other with trichrome staining. (**B**). Two red arrows indicate the rupture point in the inner and outer layers with partial dislodgement.

**Figure 3 biomedicines-11-00220-f003:**
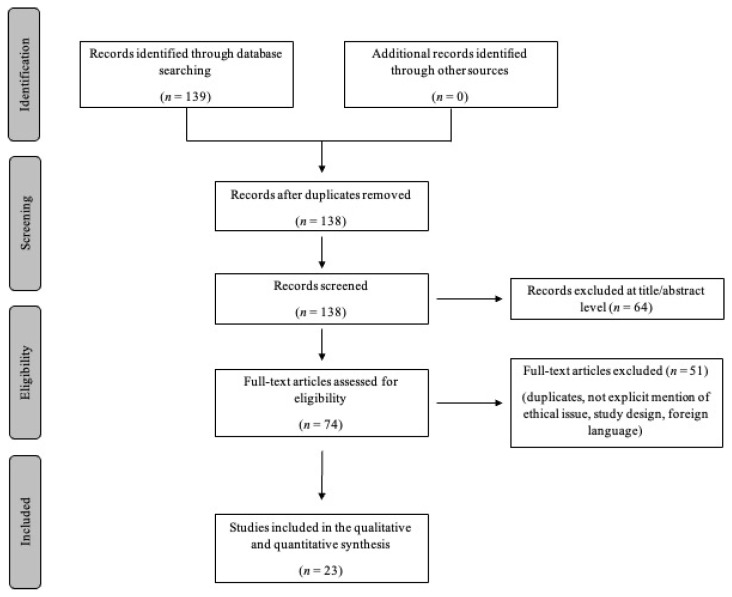
The figure shows the selection diagram of articles deemed useful for the meta-analysis. From a total of 138 articles, only 23 papers were selected.

**Table 1 biomedicines-11-00220-t001:** The table shows 23 cases of SVG aneurysmal or pseudoaneurysmal rupture. In each case, the age, sex, and time of rupture were described. A separate column listed the imaging tests useful for diagnosis and the patients’ comorbidities.

Reference	n° cases	Age (y) and Gender	Time	Rupture Modality	Outcome	Diagnostic Investigations	Autopsy	Symptoms	Description	Morbidities
Aneurysms (14)										
Riahi M et al. 1975 [6]	1	40 M	3 years	Aneurysmal dilatation and rupture of SVG	Alive	RX, Aortography	No	Asymptomatic	Vein graft aneurysm which appeared 6 months after surgery. The surgical operation consisted of an aortic valve replacement and a right coronary artery SVG.	Severe aortic insufficiency
Benchimol et al. 1975 [7]	1	62 M	4 months	Aneurysmal dilations and rupture of SVG	Alive	Angiography	No	Asymptomatic	After aorto-coronary SVG implantation, the patient underwent aortocoronary graft angiography demonstrating multiple venous aneurysmal dilations.	Diabetes Mellitus
Douglas et al. 1979 [8]	1	33 M	7 days	Rupture of mycotic aneurysm in the midportion of the SVG	Dead	NA	Yes	Fever, leukocytosis, hemorrhage	After SVG surgery, the patients developed an infective SVG aneurysm. Initially, no infection was documented. After 5 days, later tamponade secondary to the recurrence of hemorrhage was fatal.	Smoker, Coronary artery disease
Bramlet DA et al. 1982 [10]	1	46 M	8 years	Aneurysmal dilatation and rupture of SVG	Dead	Angiography	Yes	Nocturnal recurrent chest pain	Patients resulted affected by multiple coronary artery aneurysms in which a dissecting SVG aneurysm developed.	Hypertension, Smoker, Myocardial infarction, Aneurysms of coronaries
Shapeero LG et al. 1983 [11]	1	56 M	9 years	Aneurysmal dilatation and rupture of SVG	Alive	RX, CT, Cardiac catheterization, Angiography	No	Unstable Angina	The SVG mass was developed 3 months after the last radiological control. The mass compressed the main and left pulmonary arteries. Inside, a thrombotic formation was present. During this period patient suffered a heart attack.	NA
Murphy JP et al. 1986 [12]	1	65 M	14 years	Aneurysmal dilations and rupture of SVG	Alive	RX	No	Intermittent anterior pleural chest pain, Hemothorax	Reversed SV grafts had been placed to the right and the left anterior descending artery. After the SVG rupture, 1600 mL of fresh blood was evacuated from the right chest. The patient was operated on with systemic cooling.	NA
Yousem D et al. 1986 [13]	1	23 F	5 years	Rupture	Dead	RX, CT, Transthoracic echocardiogram	Yes	NA	CT revealed two large fusiform structures of low attenuation. Echocardiography showed vascular origin. Several months later, the patient died suddenly at home.	Nonspecific Vasculitis; Chronic renal failure
Forster DA et al. 1991 [9]	1	62 M	17 years	Intraoperative rupture of aneurismatic SVG	Alive	RX, CT	No	Difficulty breathing and chest pain	The patient was operated on for a thoracic mass suspicious of a cyst or teratoma. During right thoracotomy, after the removal of a clot and bleeding, it was identified as an SVG aneurysm.	NA
Steg PG et al. 1997 [14]	1	40 M	8 years	SVG rupture and false aneurysm formation	Alive	Contrast enhanced CT; Coronal spin-echo ECG-gated MRI; Aortography	No	Atypical chest pain	A sudden rupture of a thrombotic SVG developed a false aneurism. Recurrent bleedings were detected.	NA
Távora FR et al. 2007 [15]	1	39 M	10 years	Aneurysmal dilations and rupture of SVG	Dead	CT	Yes	Chest pain, hematemesis, severe respiratory distress, pleural effusion.	Admitted to the hospital with hematemesis ten years after aortocoronary bypass surgery. CT images revealed three aortocoronary SVG aneurysms but failed to detect any rupture. His subsequent death due to a rupture of the SVG aneurysm was documented at autopsy.	NA
Taguchi E et al. 2010 [16]	1	82 F	18 years	Aneurysmal dilations and rupture of SVG	Dead	CT, Coronary angiography	Yes	Shock state	The patient showed a sudden state of shock. Serological examinations were normal except for a low creatinine increase. CT revealed a mediastinal mass. The patient fell into cardiac arrest during an arteriography procedure.	Coronary artery disease
Salcedo JD et al. 2013 [17]	1	83 M	12 years	SVG aneurismatic rupture during angiography examination	Alive	Coronary angiography	No	Asymptomatic	The patient was operated on for cardiac ischemia. During the contrast injection for diagnostic coronary angiography, the graft erupted with contrast extravasation into the surrounding tissue.	Peripheric and Coronary arterial disease
Koshy GB et al. 2020 [18]	1	65 M	16 years	Aneurysmal dilations and rupture of SVG	Alive	CT, Transthoracic echocardiography, Left heart catheterization, Coronary angiography	No	Chest pain and shortness of breath	Chest pain occurred for 3 days then it was accompanied by chest pressure. Serological examinations were normal. During the angiography, extravasation of contrast was demonstrated. The patient was treated by catheterization.	Hypertension, Hyperlipidemia, Chronic obstructive pulmonary disease
Montgomery R et al. 2021 [19]	1	60 M	12 years	Aneurysmal Dilations, infection and rupture of SVG	Alive	Coronary angiography, Echocardiogram	No	Chest pain and shortness of breath, hypotension	Two months after the occurrence of Staphylococcus aureus Bacteremia, the patient developed a rupture of an infected SVG aneurysm resulting in pericardial tamponade.	Surgical site infection
Pseudoaneurysms (9)										
Rosin MD et al. 1989 [20]	1	46 M	8 years	SVG spontaneous rupture and pseudoaneurysm formation	Alive	Aortography	No	Chest pain with collapse	The patient manifested sudden hypotension. On CT examination, a mediastinal mass was evident, which was identified as a pericardial cyst. On radiological testing, bleeding near the right atrium was evident. Emergency surgery showed the presence of a pseudoaneurysm with fresh blood and thrombi in the mediastinum.	NA
Karwande SV et al. 1990 [21]	1	45 M	13 years	SVG spontaneous rupture and pseudoaneurysm formation	Alive	RX, CT	No	Chest and back pain	A mediastinal mass was discovered during radiological examinations. After the patient suffered an episode of heart attack a source of brisk bleeding was found during the operation, and it was identified as an SVG pseudoaneurism.	Hyperlipidemia; Hypertension
Werthman PE et al. 1991 [22]	1	63 M	3 years	SVG spontaneous rupture and pseudoaneurysm formation	Alive	CT, Aortography	No	Chest pain	After a heart attack episode, a mediastinal mass was discovered. After four months, the mass was discovered to be a pseudoaneurysm secondary a rupture of an SVG.	Coronary artery disease, Peripheral vascular disease
Dimitri WR et al. 1992 [23]	1	63 M	NA	Pseudoaneurysm SVG formation and rupture	Alive	CT, Aortography, Vein graft angiography	No	Massive hemorrhage and hypokinesia of the cardiac inferior segment	A SVG pseudo aneurysm caused numerous episodes of profuse intermittent bleeding through the sternum with a dehiscence of sternal wound healing.	Extensive triple vessels disease; Peripheral vascular disease
Kallis P et al. 1993 [24]	1	45 M	13 years	SVG pseudo aneurysmatic spontaneous rupture	Dead	RX, CT, Cardiac catheterization, Angiography	Yes	Dyspnea	Radiological examination showed critical stenosis of vein graft and coronaries but also a mass on the left side of the pulmonary artery. The aneurismal part was resected. Only histological examination revealed a pseudo-aneurismatic SVG rupture.	Triple vessel cardiac disease, Severe aortic valve dilatation.
Mohara J et al. 1998 [25]	1	73 F	1 month	SVG rupture and pseudoaneurysm formation	Alive	NA	No	Cardiogenic tamponade	The patient was urgently operated on for triple vessel cardiac disease. One month later angiography resulted in normal but after a few days, a posterior pseudoaneurysmal rupture was verified. The patient was urgently operated	Triple vessel cardiac disease
Puri R et al. 2009 [26]	1	61 M	13 years	SVG pseudo aneurysmatic spontaneous rupture	Alive	RX, CT, Transthoracic echocardiography, Angiography	No	Cardiogenic shock	The patient developed dyspnea but all radiological examinations resulted negative for SVG alterations. After five days echocardiography revealed cardiac tamponade. The haematoma was evacuated.	NA
Smer et al. 2015 [27]	2	91 F	NA	Pseudoaneurysm SVG formation and rupture	Alive	RX, CT, Transthoracic echocardiography	No	Hemotorax	An incidental mediastinal mass on chest X-ray with a continuous flow at the eco doppler was found. It was complicated three months later with a rupture. Conservative treatment was elected.	Advanced dementia, Hypertension, Three-vessel CABG
		80 M	2 weeks	Pseudoaneurysm SVG formation and rupture	Alive	CT, Transthoracic echocardiography	No	Chest pain, respiratory distress, peripheral edema, jugular venous distension, bibasilar crackles	The patient was operated on for a valvular replacement. After the operation, an SVG rupture with a pericardial hematoma formation developed. The aneurism was resected and the dehiscence sutured.	Severe aortic insufficiency, Two-vessel CABG

“CABG” means “coronary artery bypass graft surgery”; “SVG” means “saphenous vein graft”; “NA” means “not available”; “CT” means “computer tomography”; “TTE” means “transthoracicecography”.

**Table 2 biomedicines-11-00220-t002:** Provides a summary of the main information that can be gleaned from Table 1. Individuals were divided according to sex, age, studying their time to rupture, and outcome. The percentage was calculated based on the total number of cases (23).

	Type of Lesion		Dead	Alive	Age (y)		Rupture Time (y)	
	Aneurysm	Pseudoaneurysm			≤45	>45	≤5	>5
Male	12 (52.2)	7 (30.4)	4 (17.4)	15 (65.2)	6 (26.1)	13 (56.5)	5 (21.7)	13 (56.5)
Female	2 (8.7)	2 (8.7)	2 (8.7)	2 (8.7)	1 (4.3)	3 (13)	2 (8.7)	1 (4.3)
NA	-	-	-	-	-	-	2 (8.7)	
Total	14 (60.9)	9 (39.1)	6	17 (73.9)	7 (30.4)	16 (69.6)	7 (30.4)	14 (60.9)

**Table 3 biomedicines-11-00220-t003:** The table shows the main morbidities in the analyzed population relative to gender. Only the most frequent conditions in the general population were chosen. Absolute counts of individual pathologies were considered, and a single case may involve multiple comorbidities. In the fourth column, COPD means “chronic obstructive pulmonary disease”. In the last column, under “other vascular disease,” only aneurysms, valvopathies, and myocardiopathies were considered. Coronary artery disease or diffuse vascular disease were not considered because if patients required a graft, we implied that all had ongoing coronary artery disease.

	Diabetes Mellitus	Hypertension	Hyperlipidemia	COPD	Other Vascular Disease
Male	1	3	2	1	7
Female	-	1	-	-	5

**Table 4 biomedicines-11-00220-t004:** Subjects above and below 45 years of age were identified, and the time of aneurysm or pseudoaneurysm breakup (before or after 5 years) was determined. The percentage was calculated on the total number of male individuals [23]. In two cases, which are not counted in the table, the time to the breakup was not defined (NA)0.

Rupture Time (y)			
Age (y)	≤5	>5	NA
≤45	3 (13.4)	4 (17.4)	-
>45	4 (17.4)	10 (43.5)	2 (8.7)

**Table 5 biomedicines-11-00220-t005:** Subjects above and below 45 years of age were identified, and for each one, aneurysm or pseudoaneurysm affection was studied, determinating death or not. The percentage was calculated based on the total number of male individuals [23].

	Aneurysm		Pseudoaneurysm	
Age (y)	Alive	Dead	Alive	Dead
≤45	2 (8.7)	4 (17.4)	1 (4.3)	1 (4.3)
>45	7 (30.4)	1 (4.3)	7 (30.4)	-
Total (19)	9 (39.1)	5 (21.7)	8 (34.9)	1 (4.3)

**Table 6 biomedicines-11-00220-t006:** The table shows the total number of cases in which an autopsy was performed. The percentage was calculated based on the total number of dead subjects [23].

	Autopsy	
	Performed	Not Performed
Male	4 (17.4)	15 (65.2)
Female	2 (8.7)	2 (8.7)
Total	6 (26.1)	17 (73.9)

**Table 7 biomedicines-11-00220-t007:** The table shows the main autopsy findings in the cases analyzed. Both macroscopic and histological findings in aneurysm and pseudoaneurysm cases were analyzed.

Autoptic Data	Aneurysm (*n* = 5)	Pseudoaneurysm (*n* = 1)
Internal Examination		
Pericarditis	1	0
Hemorrhage	1	0
Graft rupture	2	1
Graft occlusion	1	1
Graft aneurysm	5	0
Graft pseudoaneurysm	0	1
Dissecting aneurysm (graft)	1	0
Coronary atherosclerosis	2	0
Hemopericardium	1	0
Hemomediastinum	2	0
Heart weight grown	2	0
Left ventricle dilatation	2	0
Lung congestion	1	0
Histology		
Heart: Polymorphonuclear infiltrate	1	0
Acute phlebitis (graft)	1	0
Coronary atherosclerosis	2	0
Graft atherosclerosis	2	1
Graft dissection	1	0
Intramural hemorrhagic dissection (graft)	1	0
Myocardial infarct	2	0
Thrombi	1	1
Hemorrhage	2	0
Fibrin	2	0
Calcification	2	0
Myocardial interstitial fibrosis	1	0
Myocyte hypertrophy	1	0
Graft occlusion	1	1
Graft aneurysm	5	0
Graft pseudoaneurysm	1	0

## Data Availability

Not applicable.

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
