# Peer review of "Early Saphenous Vein Graft Aneurysm Rupture: A Not So-Late Complication. Case Report and Comprehensive Literature Review"

_biomedicines, 2023, doi:10.3390/biomedicines11010220_

Round 1

Reviewer 1 Report

I read with interest the paper by Mezzetti et presenting a rare case of early venous graft aneurysm rupture and a review of the literature. After careful reading of the manuscript I have the following comments/questions/suggestions for the authors:

1. Title: The title should be: A rare case of early saphenous vein graft …

2. In the abstract and in the text I miss the explanation of SVG when first appeared. It should be saphenous vein graft (SVG) and then use the abbreviation SVG.

3. Introduction first line: Coronary artery bypass grafting (CABG) is the correct term.

4. “The graft connects the aorta and coronaries for myocardial revascularization” is not completely correct. First, in the case of using in situ the internal thoracic arteries, there is only coronary artery bypass without the need to attach the graft in the aorta. Second, the correct expression is that the graft bypasses the coronary artery blockage which is the result of atherosclerotic disease.

5. “The saphenous vein is often used as a venous graft because this technique has high failure rates, ranging from 10 to 25 per cent in the first 12-18 months …”. The correct meaning should be “The saphenous vein is often used as a venous graft, however, this technique …”

6. “ … the surgeons performed an immediate double bypass through a bilateral autologous SVG.” The term bilateral autologous SVG does not make sense from surgical point of view. The authors should clearly describe the two bypass which were performed. I assume one SVG graft to the LAD and one SVG graft to the OM branch of the LCx.

7. “The physicians hypothesized that myocardial hypoperfusion was caused by ostial occlusion that occurred during valve implantation”. First, the correct term is coronary artery ostial occlusion. The authors should clarify which coronary artery ostium was occluded? The left, the right or both. I assume that in this case it was occluded the left coronary artery ostium. The authors should also clarify whether the surgeon checked for coronary artery ostium occlusion after valve suturing. This is a routine checking in aortic valve replacement before aortotomy closure and the term “the physicians hypothesized such an occlusion” is problematic. Why the authors did not re-open the aortotomy to confirm the coronary ostium occlusion and probably to remove the valve and implant another one, rather than proceeding with CABG. What was the distance of the coronary artery ostia from the aortic annulus prior to aortic valve suturing. What kind of aortic valve was used in this operation?

8. “The autopsy was performed and revealed rupture of a pseudoaneurysm in the left coronary graft, which caused massive blood leakage”. First, I think it is better to write: The autopsy revealed … . Second, “rupture of a pseudoaneurysm in the left coronary graft …” does not make sense from surgical point of view. The authors should clarify which SVG graft developed pseudoaneurysm and then presented with rupture. It is obvious the rupture occurred in the pericardial space and not in the coronary artery. I would suggest a cardiac surgeon to read the manuscript and clarify the surgical procedures, because these are not well presented from surgical point of view in some cases throughout the manuscript.

9. “In coronary heart surgery, coronary heart bypass grafting (CABG) consists of creating communication in cardiac circulation between two parts of a vessel using an arterial, venous, or synthetic vessel”. This sentence is problematic. First, the correct term is CABG which means coronary artery bypass grafting and the definition has been already presented in the introduction. The term “Coronary heart surgery” is not used in the clinical practice and it can create confusion. The term “creating communication in cardiac circulation between two parts of a vessel” is also problematic and makes no sense from surgical/clinical point of view. CABG is a procedure to bypass the stenotic lesion of the coronary artery and there are aorto-coronary grafts (free arterial grafts and SVGs) or the use of internal thoracic arteries in situ with only a distal anastomosis in the coronary artery. Finally, I don’t know any case of synthetic graft in the clinical setting. If the authors are aware of such an application in the clinical setting they should provide a reference.

10. Some abbreviations (i.e. AR, AEI, AGED, LAD) are used only once and therefore there is no need to create them.

11. “It would be erroneous to continue to classify aneurysmal rupture as an exclusively late complication, although it rarely occurs within a few months or days of surgery. Certainly, aneurysmal rupture can occur early, even in the absence of infectious disease”. I disagree with this conclusion. The authors are presenting only one case of early rupture and this can not be considered as a new classification. To support my thought, I would like to remind to the authors that the CABG procedure in this case was performed under urgent conditions and there is high possibility that the quick SVG aneurysm formation was indeed due to iatrogenic manipulation. Do the surgeons mentioned anything about the quality of SVG graft during dissection and also during testing with heparinized saline prior to anastomosis.

12.  “If the injury had been secondary to graft manipulation, the rupture of the two layers would have been simultaneous, and there would have been no evidence of sloughing”. I don’t understand why a partial iatrogenic injury to the SVG can not lead to aneurysm formation after putting the graft into the arterial circulation. The authors should provide some comments regarding the use of the SVGs in the arterial circulation and its consequences.

13. “Because bypass surgery is often performed under urgent conditions, preoperative evaluation of the graft status sometimes is impossible”. This sentence is problematic from surgical point of view. First, CABG is not often performed under urgent conditions. The authors can check large series of CABG cases and see the frequency of elective, urgent and emergent CABG cases. Second in their case CABG was not a planned option but it was decided to be performed because of aortic valve replacement complication. Third, there is always intraoperative evaluation of the SVGs as I mentioned above with heparinized saline in order to confirm the integrity of the graft prior to anastomosis. Sometimes it is needed to harvest another part of the saphenous vein if the ones already prepared are of low quality or if they have parts of varicose disease. Do the surgeons detected any signs of varicose disease in the SVGs used in this case. The presence of varicose disease is potentially a risk factor for aneurysm development.

14. “The most reliable diagnostic technique for the diagnosis of aneurysmal rupture is CT or CT Angio examination. Certainly, a chest X-Ray or echocardiographic examination should be performed in the first instance. Transesophageal echocardiography retains a sensitivity and specificity of 99% and 98% for cardiac lesions”. In my opinion another diagnostic approach with very high sensitivity and specificity is coronary angiogram and more specifically angiogram of the SVGs. In this case the interventional cardiologist has also the ability to control the rupture with stent implantation and this has the potential to be a successful life-saving therapeutic approach. In this case report the authors should mention in which part of SVG was found the aneurysm: in the distal anastomosis, in the proximal anastomosis or in the main body of the graft?

Author Response

Dear reviewer, first of all, we thank you for your comments because they helped us to improve our paper and clarify the concepts we wanted to present. As suggested, we responded to your observations point by point after submitting the article to a cardiac specialist. We hope we have done a good job.

Reviewer 2 Report

Thank you for the opportunity to review the article entitled: “A rare case of early venous graft aneurysm rupture: a review of the literature.” By Mezzetti and colleagues. I thoroughly enjoyed this paper and congratulate the authors on a well-described and written paper on past cases of this condition and providing a profound review on this topic. I do have some comments though.

I appreciate the histological slices of the ruptured vein, this adds greatly to the paper even though it does not provide additional valuable clinical decision-making information.

The first three paragraphs of the discussion are not necessary, the authors should get right into the main theme and describe their results.

This is an interesting case and an interesting paper, I do not believe this will add to the current literature.

My main problem with this paper is the following, the patient had surgery and was deteriorating for an entire day before taken back to surgery, the patient was intervened too late, and as a result died. I find this a huge fault on the part of the caregivers.

Although this paper provides an interesting review on the topic (pathologically speaking), surgically I’m having serious problems accepting the care provided to the patient. 

Author Response

Dear reviewer, we thank you for your feedback. We have made some changes to the article that are marked in yellow.
We have also responded to the comments you made.
We renew our thanks and hope we have done a good job.

Round 2

Reviewer 1 Report

I reviewed the revisions and I have the following comments/suggestions/questions:

1. Since one abbreviation is explained then use only the abbreviation. This refers to CABG which is also explained in the discussion section.

2. Introduction: start without A … start the sentence as: Coronary artery bypass grafting (CABG) …

3. The right marginal artery is a branch of the RCA or of the LCx? Please clarify.

4. Occlusion of coronary ostia can NOT be a frequent occurrence, as the authors claim in replying in my previous comment #7.

5. Bypass insertion is not an accepted term. You can say: CABG is performed.

6. The authors clarified that the patient underwent a procedure that involved replacing “en bloc” the ascending aorta, aortic root and aortic valve with a bioprosthesis (Edwards Inspiris Resillia 29mm). Do the authors mean that the patient had a Bentall operation? In this case why the surgeons did not use a composite graft? Aortic root replacement means that the coronaries should be reimplanted in the synthetic graft which replaced the aortic root. In this case it is difficult to have coronary ostial obstruction. However, in this case there might be coronary artery obstruction by kinking due to inappropriate length of the reimplanted coronary artery(ies). The detailed description of the procedure which was performed by surgeons (simultaneous replacement of the ascending aorta, aortic root and aortic valve) needs further  clarifications, because in this case it is not a simple aortic valve replacement. Accordingly, the mechanism of coronary arteries obstructions is not clear if it is a result of aortic valve implantation. From surgical point of view there is now more confusion and significant issues which need clarifications in order to understand the etiology of coronary artery obstruction.

7. The mention of synthetic materials as future option for CABG is not related with the subject of this manuscript and it should be removed.

8. The fact that no iatrogenic changes were detected after SVG dissection in the macroscopic level does not preclude the presence of wall deficits in the microscopic level. High pressure of heparin solution flashing has the potential to damage the venous endothelium. In an emergency situation, such in this case, it may be such a damage of the SVG because of rushing.

Author Response

Dear Reviewer, we thank you once again for the corrections you have made to the text and your precious suggestions. We have taken steps to answer all your questions and to modify the text according to your advice. Changes are marked in yellow.

We hope we did a good job. Please see the attached file.

Reviewer 2 Report

The author have adequately responded my questions, I have no additional comments

Author Response

Dear Reviewer, we thank you once again for the feedback you gave us, and we are pleased that you enjoyed our work.

Round 3

Reviewer 1 Report

Further comments to the second revision:

1. Since the operation is a Bentall - De Bono operation the authors should describe the composite graft used. The are providing only a description for aortic valve (Edwards Inspiris Resilia 29mm). Is this valve manually sutured in a synthetic tube graft? Is this a wise decision in an emergency operation or there was lack of a composite graft?

2. How the coronaries were re-implanted in a synthetic graft which is not described?

3. Line 69: The type of bypass packed does not make sence: please correct to: the type of bypass performed was ... 

4. Line 92 correct FV to VF.

Author Response

Dear Reviewer, thank you again for giving us the opportunity to clarify what we have written.

We have included some clarifications in accordance with what you requested (highlighted in yellow).

We hope we did a good job.
